# Multi-domain Causal Structure Learning in Linear Systems

**AmirEmad Ghassami**[*]**, Negar Kiyavash**[†]**, Biwei Huang**[‡]**, Kun Zhang**[‡]
[*]Department of ECE, University of Illinois at Urbana-Champaign, Urbana, IL, USA.
[†]School of ISyE and ECE, Georgia Institute of Technology, Atlanta, GA, USA.
[‡]Department of Philosophy, Carnegie Mellon University, Pittsburgh, PA, USA.

## Abstract

We study the problem of causal structure learning in linear systems from observational data given in multiple domains, across which the causal coefficients and/or the distribution of the exogenous noises may vary. The main tool used in our approach is the principle that in a causally sufficient system, the causal modules, as well as their included parameters, change independently across domains. We first introduce our approach for finding causal direction in a system comprising two variables and propose efficient methods for identifying causal direction. Then we generalize our methods to causal structure learning in networks of variables. Most of previous work in structure learning from multi-domain data assume that certain types of invariance are held in causal modules across domains. Our approach unifies the idea in those works and generalizes to the case that there is no such invariance across the domains. Our proposed methods are generally capable of identifying causal direction from fewer than ten domains. When the invariance property holds, two domains are generally sufficient.

## 1 Introduction

Consider a system comprised of two dependent random variables $X$ and $Y$ with no latent confounders. Assuming that the dependency is due to a unidirectional causation, how can we determine which variable is the cause and which one is the effect? The golden standard for answering this question is performing controlled experiments or interventions on the variables. Randomizing or intervening on the cause variable may change the effect variable, but not vice versa [Ebe07]. The issue with this approach is that in many cases performing experiments are expensive, unethical, impossible, or even undefined. Therefore, there is a high interest in finding causal relations from purely observational data. Unfortunately, for a general system, using traditional conditional independence-based methods on purely observational data can identify a structure only up to its Markov equivalence [SGS00, Pea09]. For instance, for the aforementioned system of only two variables, the structures $X \rightarrow Y$ and $Y \rightarrow X$ (arrow indicates the causal direction), are Markov equivalent. Therefore, extra assumptions are required to make the problem well-defined.

One successful approach in the literature is to restrict the data generating model. For the case that data is generated from a structural equation model with additive noise [Bol89], if the noise is non-Gaussian [SHHK06], or if the equations are non-linear with some mild conditions [ZC06, HJM$^+$09, ZH09], the structure may be identifiable. Note that for additive noise models, in the basic case of a linear system with Gaussian noise, the causal direction is non-identifiable [SHHK06].

Another recently developing approach is to assume that the data is non-homogenous and is gathered from several different distributions [Hoo90, TP01, PBM16, ZHZ$^+$17, GSKZ17, HZZ$^+$17]. In many real-life settings, the data generating distribution may vary over time, or the dataset may be gathered from different domains and hence, not follow a single distribution. While such data is usually problematic in statistical analysis and causes restrictions on the learning power, this property can be

---

[*]Correspondence to: AmirEmad Ghassami, email: `ghassam2@illinois.edu`.

leveraged for the purpose of causal discovery, which is our focus in the paper. Moreover, it has been shown that causal modeling can be exploited to facilitate transfer learning [SJP$^+$12, ZSMW13]. This is because of the coupling relationship between causal modeling and distribution change: Causal models constrain how the data distribution may change, while distribution change exhibits such changes.

In this paper, we focus on linear systems, with observational data for the variables in the system given in multiple domains, across which the causal coefficients and/or the distribution of the exogenous noises may vary. We will present efficient approaches to exploiting such changes for causal structure learning. The proposed methods are based on the principle that although the cause and the effect variables are dependent, the mechanism that generates the cause variable changes independently from the mechanism that generates the effect variable across domains. The same principle was utilized in [ZHZ$^+$17] for exploiting non-stationary or heterogeneous data for causal structure learning. However, since that work considers a non-parametric approach, it is restricted to general independence tests among distributions, which may not have high efficiency. The remaining previous works on causal discovery from multiple domains usually assume that a certain type of invariance holds across domains [PBM16, GSKZ17]. Our approach unifies the idea in those works, and generalizes to the case that there is no such invariance across the domains. We present structure learning methods, which are generally capable of identifying causal direction from fewer than ten domains (as a special case, when the invariance property holds, two domains generally suffices).

The rest of the paper is organized as follows. We first introduce our approach for finding causal relations from multi-domain observational data in a system comprising two variables in Section 2. We propose our methods for identifying causal direction in Subsections 2.1 and 2.2. These methods are evaluated in Subsection 2.3. We generalize our method to causal structure learning in networks of variables in Section 3, where the generalized version of our methods are presented in Subsections 3.1 and 3.2 and evaluated in Subsection 3.4. We also present an approach for combining our proposed methods with standard conditional independence-based structure learning algorithms in Subsection 3.3. All the proofs are provided in the supplementary materials.

## 2 Identifying Causal Relation between Two Variables

In a general system, each variable is generated by a causal module, with conditional distribution $P(\textit{effect} \mid \textit{causes})$, which takes the direct causes of the variable as input. We assume that the system is causally sufficient, that is, the variables do not have latent confounders. Given a causal directed acyclic graph (DAG), the causal Markov condition [SGS00, Pea09], as a more formal version of Reichenbach's common cause principle [Rei91], says that any variable is independent from its non-descendants given its direct causes. The contrapositive implies that if two quantities are dependent, there must exist a causal path between them. As a consequence, when the joint distribution of a causally sufficient system changes, for any effect variable, the distribution of its direct causes, $P(\textit{causes})$, must change independently from $P(\textit{effect} \mid \textit{causes})$; otherwise, the parameter sets (or variables, from a Bayesian perspective) involved in the two modules change dependently, and there must exist a causal path connecting the parameter sets, which implies the existence of confounders and contradicts the causal sufficiency assumption. This can be viewed as a realization of the modularity property of causal systems [Pea09], and as the dynamic counterpart of the principle of independent mechanisms, which states that causal modules are algorithmically independent [DJM$^+$10], or the exogeneity property of the causal system [ZZS15]. We formalize this characteristic as follows.

**Definition 1** (Principle of Independent Changes (PIC)). *In a causally sufficient system, the causal modules, as well as their included parameters, change independently across domains.*

As mentioned in Section 1, this principle was used for causal discovery in nonparametric settings in [ZHZ$^+$17, HZZ$^+$17]. As will be explained in Subsection 2.2, the case of having an invariant causal mechanism is a special case, because a constant is independent from any other variable.

To introduce our methodology, we consider a system comprised of two dependent variables $X$ and $Y$. Observational data for variables $X$ and $Y$, or in the asymptotic case, the joint distributions of $X$ and $Y$, in $d$ domains $\mathcal{D} = \{D^{(1)}, \cdots, D^{(d)}\}$ is given. The goal is to discover the causal direction between $X$ and $Y$. We denote the ground truth cause variable and effect variable by $C$ and $E$, respectively. The relation between $C$ and $E$ is assumed to be linear. Hence, the model in domain $D^{(i)} \in \mathcal{D}$ is denoted as follows:

$$\text{domain } D^{(i)}: \qquad C = N_C^{(i)}, \qquad E = a^{(i)}C + N_E^{(i)},$$

where, $N_C^{(i)}$ and $N_E^{(i)}$ are independent exogenous noises with variances $(\sigma_C^2)^{(i)}$ and $(\sigma_E^2)^{(i)}$, respectively. Without loss of generality, we assume that the exogenous noises are zero-mean. We refer to variances of the exogenous noises and the causal coefficients as the *parameters of the system*. In general, all these three parameters can vary across the domains. For our parametric model of interest, $\sigma_C^2$ corresponds to the cause module, while $a$ and $\sigma_E^2$ correspond to the effect generation module. Therefore, PIC implies that $\sigma_C^2$ changes independently from the pair $(a, \sigma_E^2)$. Note that in general, $\sigma_E^2$ need not to be independent from $a$, as they both correspond to the mechanism generating the effect.

## 2.1 Proposed Approach

Let $\beta_{Y|X}$ be the linear regression coefficient obtained from regressing $Y$ on $X$, and let $\sigma_{Y|X}^2 = \mathrm{Var}(Y - \beta_{Y|X} \cdot X)$, i.e., the variance of the residual of regressing $Y$ on $X$. For the causal direction, we have

$$\sigma_{C|\emptyset}^2 = \sigma_C^2, \qquad \beta_{E|C} = \frac{a\sigma_C^2}{\sigma_C^2} = a, \qquad \sigma_{E|C}^2 = \sigma_E^2. \tag{1}$$

For the reverse direction, we have

$$\sigma_{E|\emptyset}^2 = a^2\sigma_C^2 + \sigma_E^2, \qquad \beta_{C|E} = \frac{\mathbb{E}[CE]}{\mathbb{E}[E^2]} = \frac{a\sigma_C^2}{a^2\sigma_C^2 + \sigma_E^2},$$

$$\sigma_{C|E}^2 = \mathrm{Var}\left(N_C - \frac{a\sigma_C^2}{a^2\sigma_C^2 + \sigma_E^2}(aN_C + N_E)\right) = \frac{\sigma_C^2\sigma_E^2}{a^2\sigma_C^2 + \sigma_E^2}. \tag{2}$$

We will utilize the change information across domains to find the causal direction as follows. For any parameter $\gamma \in \{\sigma_{C|\emptyset}^2, \beta_{E|C}, \sigma_{E|C}^2, \sigma_{E|\emptyset}^2, \beta_{C|E}, \sigma_{C|E}^2\}$, let $\gamma^{(i)}$ denote the value of this parameter in domain $D^{(i)}$, $1 \leq i \leq d$. Consider $\{\gamma^{(1)}, \cdots, \gamma^{(d)}\}$ as samples from random variable $\gamma$. As stated earlier, according to PIC, $\sigma_{C|\emptyset}^2 = \sigma_C^2$ is independent form $(\beta_{E|C}, \sigma_{E|C}^2) = (a, \sigma_E^2)$, while as we can see from the expressions in (2), such independence does not hold in general in the reverse direction. For instance, if $a$ and $\sigma_E^2$ are both fixed, an increase in $\sigma_{E|\emptyset}^2$ always leads to an increase in $\beta_{C|E}$ and $\sigma_{C|E}^2$. Therefore, we propose our causal discovery method as follows. To test whether $X$ is the cause of $Y$, we test the independence between $\sigma_{X|\emptyset}^2$ and $(\beta_{Y|X}, \sigma_{Y|X}^2)$. If $\sigma_{X|\emptyset}^2$ and $(\beta_{Y|X}, \sigma_{Y|X}^2)$ are independent but the counterpart in the reverse direction is not, $X$ is considered as the cause variable and $Y$ the effect variable. More specifically, for testing if $X$ is the cause of $Y$, let $\Gamma_{X \to Y} = \{|\beta_{Y|X}|, \sigma_{Y|X}^2\}$, and define the dependence measure

$$\mathcal{T}_{X \to Y}(\mathcal{D}) := \sum_{\gamma \in \Gamma_{X \to Y}} \mathcal{I}(\gamma, \sigma_{X|\emptyset}^2), \tag{3}$$

where any standard non-parametric measure of dependence $\mathcal{I}(\cdot, \cdot)$, such as mutual information, can be used. Therefore, for inferring the causal relation between $X$ and $Y$, we calculate $\mathcal{T}_{X \to Y}(\mathcal{D})$ and $\mathcal{T}_{Y \to X}(\mathcal{D})$ and pick the direction which has the smaller value, i.e., $\arg\min_{\pi \in \{X \to Y, Y \to X\}} \mathcal{T}_\pi(\mathcal{D})$, as the correct causal direction. Alternatively, one can use test of statistical independence, such as the kernel-based one [GFT$^+$08] to infer the direction.

### 2.1.1 Identical Boundaries Method

Although checking for independence is sufficient for discovering causal relation, in general performing a non-parametric independence test may not be efficient. This may be specially problematic as in many applications the number of domains is small. In this subsection, we show that the parametric structure of our model can be exploited to devise an efficient independence test.

The main idea is as follows. Consider two bounded random variables $\gamma$ and $\tilde{\gamma}$. We refer to the minimum and maximum of $\tilde{\gamma}$ as the boundaries of this variable. If $\gamma$ and $\tilde{\gamma}$ are independent, then $\gamma$ will have the same distribution conditioned on $\tilde{\gamma} = \tilde{\gamma}_{min}$ and $\tilde{\gamma} = \tilde{\gamma}_{max}$, i.e., $\gamma$ will have identical distributions on the boundaries of $\tilde{\gamma}$. Specifically, it will have same expected values on the boundaries of $\tilde{\gamma}$. On the other hand, if $\gamma$ and $\tilde{\gamma}$ are dependent, the expected value of $\gamma$ on the boundaries of $\tilde{\gamma}$ are not necessarily different. However, we will show that if $X \to Y$ is not the causal direction and $\beta_{Y|X}$ (and $\sigma_{Y|X}^2$) is dependent on $\sigma_{X|\emptyset}^2$, due to the specific parametric structure of our model, the expected value of $\beta_{Y|X}$ (and $\sigma_{Y|X}^2$) on the boundaries of $\sigma_{X|\emptyset}^2$ will not be identical.

For any parameter of interest, we denote its minimum and its maximum value in the dataset with subscripts $\hat{m}$ and $\hat{M}$, respectively. For inferring if $X$ is the cause of $Y$, let

$$\mathcal{I}^{IB}(\gamma, \sigma_{X|\emptyset}^2) := \left| \mathbb{E}[\log\gamma \mid_{\log(\sigma_{X|\emptyset}^2) = (\log(\sigma_{X|\emptyset}^2))_{\hat{M}}}] - \mathbb{E}[\log\gamma \mid_{\log(\sigma_{X|\emptyset}^2) = (\log(\sigma_{X|\emptyset}^2))_{\hat{m}}}] \right|,$$

and according to (3), we define the causal order indicator $\mathcal{T}_{X \to Y}^{IB}(\mathcal{D}) := \sum_{\gamma \in \Gamma_{X \to Y}} \mathcal{I}^{IB}(\gamma, \sigma_{X|\emptyset}^2)$, where *IB* stands for identical boundaries. We have the following result for identifiability.

**Theorem 1.** *For dataset* $\mathcal{D} = \{D^{(1)}, \cdots, D^{(d)}\}$, *as* $d \to \infty$, $\mathcal{T}_{C \to E}^{IB}(\mathcal{D}) \to 0$, *and* $\mathcal{T}_{E \to C}^{IB}(\mathcal{D}) \to c$, *for some positive value c which is bounded away from 0.*

Using Theorem 1, to see the causal relation between $X$ and $Y$, we calculate $\mathcal{T}_{X \to Y}^{IB}(\mathcal{D})$ and $\mathcal{T}_{Y \to X}^{IB}(\mathcal{D})$ and pick the direction which has the smaller value, i.e., $\arg\min_{\pi \in \{X \to Y, Y \to X\}} \mathcal{T}_{\pi}^{IB}(\mathcal{D})$, as the true causal direction. We term this approach the *Identical Boundaries* (IB) method.

Note that in the IB method we only perform first-order statistical test (i.e., regarding the mean) on the boundaries. Clearly, the idea can be extended to performing higher-order tests as well. We have provided the required formulation for the extension to second-order test in the supplementary materials.

## 2.2   Minimal Changes Method

One may perform causal discovery from multiple domains by assuming certain invariances in the parameters across domains. More specifically, consider a pair of domains $\{D^{(i)}, D^{(j)}\}$. In order to find the causal direction, the authors of [GSKZ17] consider the particular case that $a^{(i)} = a^{(j)}$ and either $(\sigma_C^2)^{(i)} \neq (\sigma_C^2)^{(j)}$ or $(\sigma_E^2)^{(i)} \neq (\sigma_E^2)^{(j)}$. In this case, $\beta_{E|C}^{(i)} = \beta_{E|C}^{(j)}$, and we have $\beta_{C|E}^{(i)} = \beta_{C|E}^{(j)}$ if and only if $\sigma_C^{(i)}/\sigma_C^{(j)} = \sigma_E^{(i)}/\sigma_E^{(j)}$. Therefore, comparing the coefficients in both directions, the causal direction is identifiable if $\sigma_C^{(i)}/\sigma_C^{(j)} \neq \sigma_E^{(i)}/\sigma_E^{(j)}$. Note that if either of the variances does not change, this condition always holds. In [PBM16], the authors assume that $(\sigma_E^2)^{(i)} = (\sigma_E^2)^{(j)}$. Therefore, for variable $E$ and any set $S$, if $(\sigma_{E|S}^2)^{(i)} \neq (\sigma_{E|S}^2)^{(j)}$, then $S$ is not the parent set of $E$. Similarly, the authors of [WSBU18] consider the case that $a^{(i)} \neq a^{(j)}$ and either $\sigma_C^{(i)} = \sigma_C^{(j)}$, or $\sigma_E^{(i)} = \sigma_E^{(j)}$. Therefore, under this assumption, in this case for variable $X \in \{C, E\}$ and set $S \subseteq \{C, E\} \setminus X$, if $(\sigma_{X|S}^2)^{(i)} = (\sigma_{X|S}^2)^{(j)}$, then $S$ is the parent set of $X$.

We note that invariance is a special case of the condition of independent changes, as a constant is independent from any variable. Therefore, our idea introduced in Subsection 2.1 can be applied to the case of existence of invariant parameter across domains. In this case, two domains are generally sufficient to identify the causal direction. Therefore, we give a unification and generalization of the perspectives of the aforementioned previous works. The assumptions in these works can be seen as particular realizations of the faithfulness assumption [SGS00], which is required for our proposed approach as well:

**Assumption 1.** *For any pair of domains* $D^{(i)}$ *and* $D^{(j)}$, *if* $(\sigma_{E|\emptyset}^2)^{(i)} = (\sigma_{E|\emptyset}^2)^{(j)}$, *or* $(\beta_{C|E})^{(i)} = (\beta_{C|E})^{(j)}$, *or* $(\sigma_{C|E}^2)^{(i)} = (\sigma_{C|E}^2)^{(j)}$, *then all parameters of the system are unchanged across* $D^{(i)}$ *and* $D^{(j)}$, *i.e.,* $(\sigma_C^2)^{(i)} = (\sigma_C^2)^{(j)}$, *and* $a^{(i)} = a^{(j)}$, *and* $(\sigma_E^2)^{(i)} = (\sigma_E^2)^{(j)}$.

Assumption 1 is mild in the sense that it only rules out a 2-dimensional subspace of a 3-dimensional space. Therefore, considering Lebesgue measure on the 3-dimensional space, we are only ruling out a measure-zero subset.

Since invariance is a special case of independent changes, based on PIC, change in one causal module does not force any changes in another causal module, i.e., a change in, say, $\sigma_{C|\emptyset}^2$, will not enforce any changes on $\beta_{E|C}$ or $\sigma_{E|C}^2$. However, in the reverse direction, as it can be seen from equations in (1) and (2), if any of the variables $a$, $\sigma_C^2$, and $\sigma_E^2$ varies across two domains, by Assumption 1, all three variables $\sigma_{E|\emptyset}^2$, $\beta_{C|E}$, and $\sigma_{C|E}^2$ will change. Based on this observation, we propose the following principle for causal discovery, which is the counterpart of PIC for the case of invariance.

**Definition 2** (Principle of Minimal Changes (PMC)). *Suppose Assumption 1 holds. Compared to the direction from effect to cause, fewer or an equal number of changes are required in the causal direction to explain the variation in the joint distribution.*

Therefore, for testing whether variable $X$ is the cause of variable $Y$, we propose the following method. Let $\Gamma'_{X \to Y} := \{\sigma_{X|\emptyset}^2, |\beta_{Y|X}|, \sigma_{Y|X}^2\}$. We denote a member of $\Gamma'_{X \to Y}$ by $\gamma$, and its realization in domain $\mathcal{D}^{(i)}$ by $\gamma^{(i)}$. For any pair of domains $\{D^{(i)}, D^{(j)}\}$, let $Q_{X \to Y}^{(i,j)} := \sum_{\gamma \in \Gamma'_{X \to Y}} \mathbb{1}[\log \gamma^{(i)} \neq \log \gamma^{(j)}]$. This quantity counts the number of members of $\Gamma'_{X \to Y}$ that

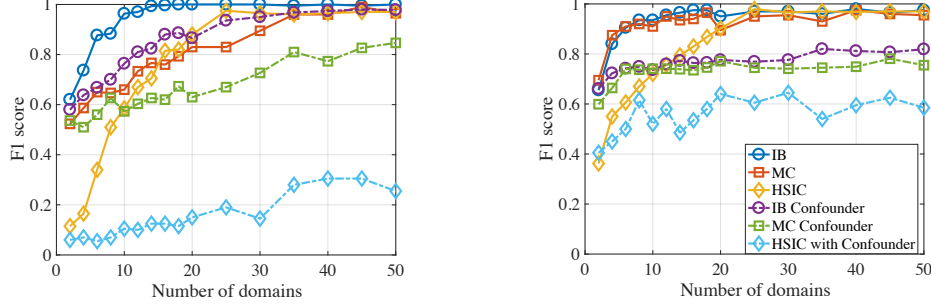

Figure 1: F1 score versus number of domains for model 1 on the left and model 2 on the right.

vary across domains $\{D^{(i)}, D^{(j)}\}$. We define the causal direction indicator

$$\mathcal{T}_{X \to Y}^{MC}(\mathcal{D}) := \sum_{1 \leq i < j \leq d} \mathbb{1}[X \to Y \in \arg \min_{\pi \in \{X \to Y, Y \to X\}} Q_\pi^{(i,j)}],$$

where *MC* stands for minimal changes. Under Assumption 1, we have the following result.

**Theorem 2.** *For a given dataset $\mathcal{D}$, we have $\mathcal{T}_{C \to E}^{MC}(\mathcal{D}) \geq \mathcal{T}_{E \to C}^{MC}(\mathcal{D})$. The inequality is strict if there exists a pair of domains across which at least one and at most two of the parameters $\sigma_C^2$, $a$, and $\sigma_E^2$ varies.*

Using Theorem 2, for testing the causal relation between $X$ and $Y$, we calculate $\mathcal{T}_{X \to Y}^{MC}(\mathcal{D})$ and $\mathcal{T}_{Y \to X}^{MC}(\mathcal{D})$ and pick the direction that has the larger value, i.e., $\arg \max_{\pi \in \{X \to Y, Y \to X\}} \mathcal{T}_\pi^{MC}(\mathcal{D})$, as the true causal direction. We term this approach the *Minimal Changes* (MC) method.

### 2.3 Evaluation

We consider two models for generating the parameters of the system. In the first model, the variances of the noises and the causal coefficients follow the distributions *Unif*([1,3]) and *Unif*([-3,-0.5] $\cup$ [0.5,3]), respectively. In the second model, with equal probability, they either follow the aforementioned distributions, or are equal to a fixed value. The number of samples in each domain is $10^3$. We have used the state-of-the-art HISC test [GFT$^+$08] as our non-parametric independence test. The performance of the proposed methods are depicted in Figure 1. We have depicted the F1 score for each case. As seen from the F1 score, the IB method performs better than the MC method in the first model. However, if in an application we know that the parameters are not likely to change much (as in model 2), the MC method also has high performance. Moreover, for the case of limited number of domains, the IB method significantly outperforms using HSIC test. We also tested our proposed methods when a latent confounder was present in the system. As could be seen in Figure 1, IB is generally more robust to the presence of latent confounder.

## 3 Causal Discovery with More than Two Variables

In this section, we extend the proposed methods in Section 2 to the case with more than two variables. We consider a causal DAG $\mathcal{G} = (V, A)$ over $p$ variables of a causally sufficient system, where $V = \{X_1, \cdots, X_p\}$ is the set of variables of the system, and $A$ denotes the set of directed edges, i.e., each element of $A$ is an ordered pair of elements of $V$. Observational data for variables in $d$ domains $\mathcal{D} = \{D^{(1)}, \cdots, D^{(d)}\}$ is given. Define the random vector $X := (X_1, \cdots, X_p)^\top$. We consider the following linear structural equation model (SEM):

$$X = B^\top X + N, \tag{4}$$

where $B$ is the matrix of coefficients of edges, with $B_{ij}$ denoting the coefficient from variable $X_i$ to $X_j$. If $B_{ij} \neq 0$, $X_i$ is a parent of $X_j$ and $X_j$ is a child of $X_i$. We denote the parent set and children set of $X$ by $Pa(X)$ and $Ch(X)$, respectively. A variable is called *source* variable if it does not have any parents, and *sink* variable if it does not have any children. Without loss of generality, we assume that $B$ is a strictly upper triangular matrix. Since the underlying structure is DAG, rows and columns of $B$ can be permuted for this condition to be satisfied. $N = (N_1, \cdots, N_p)^\top$ is the vector of exogenous noises. We denote the variance of $N_i$ by $\sigma_i^2$, and assume that elements of $N$ are independent and without loss of generality, we assume that they are zero-mean variables. Also, we define $\Omega$ as the $p \times p$ diagonal matrix with values $\sigma_1^2, \cdots, \sigma_p^2$ on the diagonal.

**Algorithm 1** Multi-domain Causal Structure Learning.

---

**Input:** Data from variables $V$ in domains $\mathcal{D}$, initial order $\pi_t$ on $V$. **Initiation:** $\hat{\pi} = \emptyset$.
**while** $|\pi_t| \neq 0$ **do**
    **for** $X \in \pi_t$ **do**
        $\Gamma_{\pi_{X,-1}(n)} = \{|(\hat{B}_{\pi_{X,-1}})_{m,n}|, (\hat{\Omega}_{\pi_{X,-1}})_{n,n}; 1 \leq m \leq n\}, 1 \leq n \leq |\pi_t|$.
        $\mathcal{T}_{X,-1}(\mathcal{D}) = \sum_{\gamma \in \Gamma_{\pi_{X,-1}(n)}} \sum_{k=1}^{n-1} \sum_{\tilde{\gamma} \in \Gamma_{\pi_{X,-1}(k)}} \mathcal{I}(\gamma, \tilde{\gamma})$.

        **For the IB method:** ─────────────────────────────
        $\mathcal{T}_{X,-1}^{IB}(\mathcal{D}) = \sum_{\gamma \in \Gamma_{\pi_{X,-1}(n)}} \sum_{k=1}^{n-1} \sum_{\tilde{\gamma} \in \Gamma_{\pi_{X,-1}(k)}} |\mathbb{E}[\log \gamma \,|_{\log \tilde{\gamma} = (\log \tilde{\gamma})_{\hat{M}}}]$
             $-\mathbb{E}[\log \gamma \,|_{\log \tilde{\gamma} = (\log \tilde{\gamma})_{\hat{m}}}]|$.

        ─────────────────────────────────────────
    **end for**
    $X_{last} = \arg\min_{X \in \pi_t} \mathcal{T}_{X,-1}^{IB}(\mathcal{D})$.
    $\hat{\pi} = \text{concatenate}(X_{last}, \hat{\pi})$, Remove $X_{last}$ from $\pi_t$.
**end while**
**Output:** $\hat{\pi}$.

---

We define an *ordering* of the variables as a permutation of the variables of the system. An ordering on a set of variables and a DAG on those variables are consistent if in the ordering, every variable comes after its parents and before its children. Note that given an undirected graph, an ordering determines the direction of all the edges of the graph uniquely; however, there may be more than one ordering consistent with a given DAG. For example, for the DAG $W \to X \to Y \leftarrow Z$, orderings $(W, Z, X, Y)$, $(W, X, Z, Y)$, and $(Z, W, X, Y)$ are consistent, but the ordering $(W, X, Y, Z)$ is not consistent with the DAG, as it contradicts with the direction of the edge between $Y$ and $Z$.

**Definition 3** (Causal Order). *An ordering on the variables is called causal if it is consistent with the ground truth causal DAG.*

Since the skeleton of the causal DAG can be identified from basic conditional independence tests, the main challenge in causal structure learning is to find a causal order. In the following we present our approaches to estimate a causal order on the variables of the system.

### 3.1 Proposed Approach

Suppose a candidate for the causal order on the variables, denoted by $\pi$, is given. In order to generalize our methodology to the network case, for each domain, we need to first estimate the regression coefficients and exogenous noise variances of each variable $X_i$ on all the variables $X_j$ with $\pi^{-1}(X_j) < \pi^{-1}(X_i)$, i.e., all the variables, which proceed $X_i$ in the given order. For the given order $\pi$ on variables, for $1 \leq m \leq p$, let $S_m := (\pi(1), \cdots, \pi(m-1))^\top$. We denote the estimated regression coefficients and exogenous noise variances for the given ordering $\pi$ by $\hat{B}_\pi$ and $\hat{\Omega}_\pi$, respectively, where $(\hat{B}_\pi)_{(1:m-1),m} = \beta_{\pi(m)|S_m}$, and $(\hat{\Omega}_\pi)_{m,m} = \sigma^2_{\pi(m)|S_m}$, and zero elsewhere.

Note that if $\pi$ is a causal order, then $\hat{B}_\pi = B$ and $\hat{\Omega}_\pi = \Omega$ up to permutation. Standard regression calculation can be used for obtaining $\hat{B}_\pi$ and $\hat{\Omega}_\pi$. We additionally propose an efficient method for estimating the regression coefficients and noise variances, provided in the Supplementary Materials. Our proposed method also makes a connection between precision matrix and adjacency matrix of a Bayesian network.

We will utilize the change information across domains to find a causal order as follows. Consider matrix $M = B + \Omega$. According to PIC, elements in each column of $M$ should be jointly independent from elements in any other column, as they correspond to distinct causal modules. Therefore, we can set a metric for measuring dependencies, and orders that obtain the minimum value are causal orders. More specifically, for a given order $\pi$ on variables, let $\hat{B}_\pi$ and $\hat{\Omega}_\pi$ be the outputs of regression. A naive way of applying the idea in Section 2 to networks is as follows: For $1 \leq n \leq p$, let $\Gamma_{\pi(n)} := \{|(\hat{B}_\pi)_{m,n}|, (\hat{\Omega}_\pi)_{n,n}; 1 \leq m \leq n\}$, and define $Q_{\pi(n)} := \sum_{\gamma \in \Gamma_{\pi(n)}} \sum_{k=1}^{n-1} \sum_{\tilde{\gamma} \in \Gamma_{\pi(k)}} \mathcal{I}(\gamma, \tilde{\gamma})$, where $\mathcal{I}(\cdot, \cdot)$ is any standard measure for dependence. We define the causal order indicator for exhaustive search $\mathcal{T}_\pi^e(\mathcal{D}) := \sum_{n=2}^p Q_{\pi(n)}$. Hence, one can estimate the causal order as $\hat{\pi} = \arg\min_\pi \mathcal{T}_\pi^e(\mathcal{D})$. Therefore, in low dimensions, the causal order can be found by exhaustive search over all orders. However, for large dimensions, this is infeasible, as the number of orders increases super-exponentially

---

**Algorithm 2** MC Causal Structure Learning.

---

**Input:** Data from variables $V$ in domains $\mathcal{D}$, initial order $\pi_t$ on $V$.    **Initiation:** $\hat{\pi} = \emptyset$.

**while** $|\pi_t| \neq 0$ **do**

    **for** $X \in \pi_t$ **do**

        Define $\Pi_X = \{\pi_t, \pi_{X,-1}, \pi_{X,-2}\}$.

        $\Gamma'_\pi = \{|(\hat{B}_\pi)_{m,n}|, (\hat{\Omega}_\pi)_{m,m}; 1 \leq m \leq n \leq |\pi_t|\}, \pi \in \Pi_X$.

        $Q_\pi^{(i,j)} = \sum_{\gamma \in \Gamma'_\pi} \mathbb{1}[\log \gamma^{(i)} \neq \log \gamma^{(j)}], \pi \in \Pi_X, 1 \leq i < j \leq d$.

        $\mathcal{T}_\pi^{MC}(\mathcal{D}) = \sum_{1 \leq i < j \leq d} \mathbb{1}[\pi \in \arg\min_{\pi' \in \Pi_X} Q_{\pi'}^{(i,j)}], \pi \in \Pi_X$.

        Update $\pi_t = \arg\max_{\pi \in \Pi_X} \mathcal{T}_\pi^{MC}(\mathcal{D})$.

    **end for**

    $\hat{\pi} = \text{concatenate}(\pi_t(-1), \hat{\pi})$, Remove $\pi_t(-1)$ from $\pi_t$.

**end while**

**Output:** $\hat{\pi}$.

---

with the number of variables. Therefore, we propose an alternative method for learning the causal structure.

The pseudo-code of the proposed approach is presented in Algorithm 1. The main idea is that in each round, we find one variable which is the last in the causal order and remove it from the list, until all the variables are ordered. The algorithm starts with a random initial order $\pi_t$ on all variables. In each round, for each variable $X \in \pi_t$, quantity $\mathcal{T}_{X,-1}(\mathcal{D})$, given in line 5 of the algorithm, is calculated, which shows the amount of dependency between parameters of the causal module of $X$ and all the other estimated parameters when $X$ is moved to the last position in $\pi_t$. After calculating the quantity $\mathcal{T}_{X,-1}(\mathcal{D})$ for all variables in $\pi_t$, the variable $X_{last}$ that has the lowest value for this quantity is concatenated to the left side of our estimated order $\hat{\pi}$, and is removed from $\pi_t$. This procedure is continued until all the variables are moved to $\hat{\pi}$.

### 3.1.1 IB Causal Structure Learning

In this subsection, we present an efficient generalization of the IB methods. For the exhaustive version of IB, we define $Q_{\pi(n)}^{IB} := \sum_{\gamma \in \Gamma_{\pi(n)}} \sum_{k=1}^{n-1} \sum_{\tilde{\gamma} \in \Gamma_{\pi(k)}} \left| \mathbb{E}[\log \gamma \mid_{\log \tilde{\gamma} = (\log \tilde{\gamma})_{\hat{M}}}] - \mathbb{E}[\log \gamma \mid_{\log \tilde{\gamma} = (\log \tilde{\gamma})_{\hat{m}}}] \right|$. We define the IB causal order indicator for exhaustive search $\mathcal{T}_\pi^{eIB}(\mathcal{D}) := \sum_{n=2}^{p} Q_{\pi(n)}^{IB}$. This gives us the following result, which is similar to Theorem 1.

**Theorem 3.** *Let $\pi^*$ be a causal and $\pi'$ be a non-causal order. For a given dataset $\mathcal{D}$, as $d \to \infty$, $\mathcal{T}_{\pi^*}^{eIB}(\mathcal{D}) \to 0$, and $\mathcal{T}_{\pi'}^{eIB}(\mathcal{D}) \to c$, for some value $c$, which is bounded away from $0$.*

Using Theorem 3, one can estimate the causal order as $\hat{\pi} = \arg\min_\pi \mathcal{T}_\pi^{eIB}(\mathcal{D})$. Therefore, in low dimensions, the causal order can be found by exhaustive search over all orders. However, as mentioned earlier for large dimensions, this is infeasible. Therefore, we need to have an alternative search approach similar to the version for non-parametric version, except that in each round, for each variable $X \in \pi_t$, quantity $\mathcal{T}_{X,-1}^{IB}(\mathcal{D})$, given in line 6 of Algorithm 1, is calculated.

**Theorem 4.** *In any round of Algorithm 1, let $X_s$ be a sink variable, and $X_v$ be a non-sink variable in the sub-graph induced on $\pi_t$. Then as $d \to \infty$, $\mathcal{T}_{X_s,-1}^{IB}(\mathcal{D}) \to 0$ and $\mathcal{T}_{X_v,-1}^{IB}(\mathcal{D}) \to c$, for some value $c$, which is bounded away from $0$.*

### 3.2 MC Causal Structure Learning

In order to generalize the MC method to the network case, we need the following assumption, which is a counterpart of Assumption 1 in the multiple variable case.

**Assumption 2.** *For any pair of domains $D^{(i)}$, $D^{(j)}$, for any variable $X$ and set $S$, if $(\sigma_{X|S}^2)^{(i)} = (\sigma_{X|S}^2)^{(j)}$ (or $[(\beta_{X|S})^{(i)}]_k = [(\beta_{X|S})^{(j)}]_k$), then the value of all the parameters of the system involved in the expression of $\sigma_{X|S}^2$ (or $[(\beta_{X|S})]_k$) are equal in the two domains.*

Roughly speaking, Assumption 2 for the linear SEM states that the parameters of the model should not have been designed in a way that they cancel each other out on correlations.

For a given order $\pi$, let $\hat{B}_\pi$ and $\hat{\Omega}_\pi$ be the output of regressions. Similar to Subsection 3.1, a naive way to apply the MC method to a network is as follows: Let $\Gamma'_\pi := \{|(\hat{B}_\pi)_{m,n}|, (\hat{\Omega}_\pi)_{m,m}; 1 \leq$

$m \le n \le p\}$, and for any pair of domains $\{D^{(i)}, D^{(j)}\}$, let $Q_\pi^{(i,j)} := \sum_{\gamma \in \Gamma'_\pi} \mathbb{1}[\log \gamma^{(i)} \ne \log \gamma^{(j)}]$. We define the MC causal order indicator for exhaustive search $\mathcal{T}_\pi^{eMC}(\mathcal{D}) := \sum_{1 \le i < j \le d} \mathbb{1}[\pi \in \arg\min_{\pi'} Q_{\pi'}^{(i,j)}]$. This gives us the following result similar to Theorem 2.

**Theorem 5.** *Let $\pi^*$ be a causal and $\pi'$ be a non-causal order. For a given dataset $\mathcal{D}$, we have $\mathcal{T}_{\pi^*}^{eMC}(\mathcal{D}) \ge \mathcal{T}_{\pi'}^{eMC}(\mathcal{D})$. Also, there exists two parameters of the system $\gamma_1$ and $\gamma_2$ such that if there exists two domains $D^{(i)}$, $D^{(j)}$ with $\gamma_1^{(i)} = \gamma_1^{(j)}$ and $\gamma_2^{(i)} \ne \gamma_2^{(j)}$, then $\mathcal{T}_{\pi^*}^{eMC}(\mathcal{D}) > \mathcal{T}_{\pi'}^{eMC}(\mathcal{D})$.*

Using Theorem 5, one can estimate the causal order as $\hat{\pi} = \arg\max_\pi \mathcal{T}_\pi^{eMC}(\mathcal{D})$. Therefore, in low dimensions, the causal order can be found by exhaustive search over all orders. However, for large dimensions, this is infeasible. Therefore, we propose the following alternative method for applying the MC method to networks.

The pseudo-code of the proposed approach is presented in Algorithm 2. The main idea is that in each round, we find one variable with lowest causal order and remove it from the list, until all the variables are ordered. The algorithm starts with a random initial order $\pi_t$ on all variables. In each round, for each variable $X$ it forms 3 orders in set $\Pi_X$. $\pi_t$ is the initial order, $\pi_{X,-1}$ is the same as $\pi_t$, but $X$ is moved to the last position, and $\pi_{X,-2}$ is the same as $\pi_t$, but $X$ is moved to one before the last position in the order.[2] The algorithm then calculates the quantity $\mathcal{T}_\pi^{MC}(\mathcal{D})$, given in line 7 of the pseudo-code, for each of the three orders in $\Pi_X$, and updates $\pi_t$ to the element of $\Pi_X$ that has the maximum value for this quantity. In the case of tie, we prioritize the orders as follows: $\pi_t > \pi_{X,-2} > \pi_{X,-1}$. This prioritization guarantees that after performing the aforementioned update of $\pi_t$ for all variables, the last variable in $\pi_t$, i.e., $\pi_t(-1)$, will be a sink variable, in the subgraph induced on variables in $\pi_t$. We concatenate $\pi_t(-1)$ to the left side of our estimated order $\hat{\pi}$, remove it from $\pi_t$, and continue to the next round until all the variables are moved to $\hat{\pi}$.

**Theorem 6.** *In each round of Algorithm 2, if $X_s$ is a sink variable, then for all $\pi \in \Pi_{X_s}$, $\mathcal{T}_{\pi_{X_s,-1}}^{MC}(\mathcal{D}) \ge \mathcal{T}_\pi^{MC}(\mathcal{D})$. Also, for any of $X_s$'s parents, $X_v$, if there exists a pair of domains across which at least one and at most two of variables $Var(X_v)$, $B_{v,s}$, $\sigma_s^2$ varies, then at the end of round, $\pi_t(-1)$ will be a sink variable.*

**Remark 1.** *Finding independence in Algorithm 1 and invariance in Algorithm 2 can also be done from top to bottom, similar to the approach used in [SIS$^+$11]. That is, in each round we can also find a variable with highest causal order as well.*

### 3.3 Combining Proposed Methods with Conditional Independence-based Algorithms

One can also run our proposed methods after applying any standard conditional independence-based algorithms, such as PC [SGS00] or GES [Chi02], to the data from some or all domains. Therefore, our proposed approach learns beyond the Markov equivalence class. The approach for combining our methods with standard conditional independence-based algorithms would be as follows.

1. We run the observational algorithm on the data from all domains (or its subset, if the sample size is too big) to learn the essential graph (also known as CPDAG) of the underlying DAG.
2. We note that an essential graph is a chain graph. For each chain component $C$, we denote its vertices by $V(C)$ and denote the set of parents of $V(C)$ by $Pa(V(C))$, and define $V_C = V(C) \cup Pa(V(C))$. Note that a variable cannot have both ingoing and outgoing edges from variables in $V(C)$, otherwise we will have a partially directed cycle, which is not allowed in a chain graph [AMP$^+$97].
3. We apply our methods on each chain component separately and use the data corresponding to $V_C$ as the input. We need to include $Pa(V(C))$ in $V_C$ to ensure that the variables under consideration do not have any latent confounders.
4. In $V_C$ we locate variables in $Pa(V(C))$ at the beginning of the order and use our methods to find the order on the remaining variables of $V_C$, i.e., on $V(C)$.
5. We combine the orders obtained from chain components.

Note that with sufficiently many changing domains, our proposed methods can learn the whole causal structure regardless of utilizing any conditional independence-based algorithm.

### 3.4 Evaluation

We considered model 1 described in Subsection 2.3 for generating the parameters of the system, with the number of generated samples in each domain equal to $10^3$. After identifying the causal ordering,

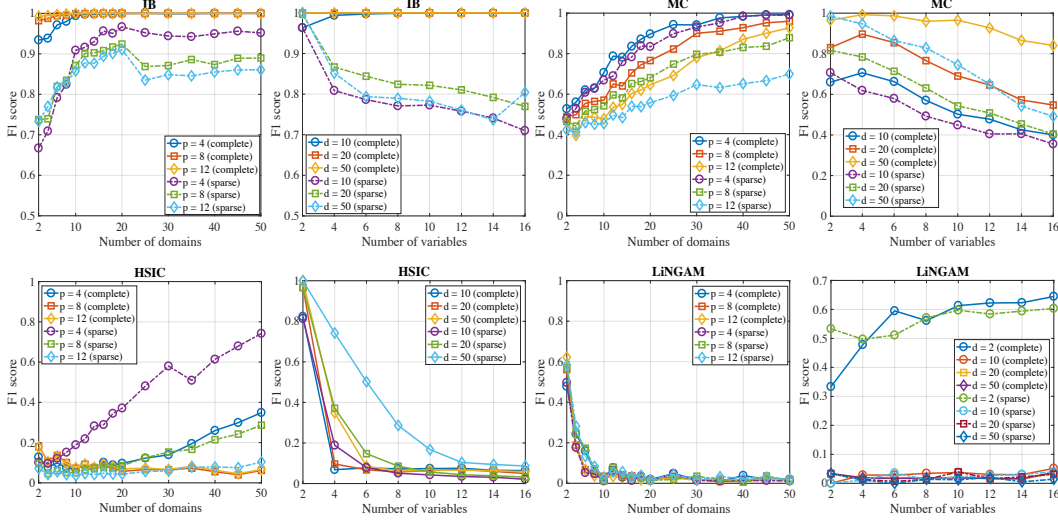

Figure 2: F1 score versus number of domains and number of variables.

we then estimate the causal coefficients $B$ on each domain separately. We set a threshold $\alpha = 0.1$ on $B$ from each domain; if $|B_{ij}|$ is larger than $\alpha$, then there is an edge from $X_i$ to $X_j$. Then if an edge appears in more than 80% of all domains, we take this edge in the final graph. The results are show in Figure 2. All experiments are performed either on complete graphs or on sparse graph generated from Erdos-Renyi model with parameter 0.3. In general, we observed better performance on denser graphs. This is expected as having more parameters helps us in predicting the order. The IB and MC methods both showed high performance in our simulations, while the performance of non-parametric HSIC test was in general not acceptable. We also compared the performance with LiNGAM Algorithm [SHHK06]. To do so, we applied LiNGAM algorithm to the pooled data of all domains. As explained in [ZHZ+17], LiNAGM failed to perform well on our multi-domain data.

**fMRI hippocampus data:** We applied our methods to fMRI hippocampus dataset [PL], which contains signals from six separate brain regions: perirhinal cortex (PRC), parahippocampal cortex (PHC), entorhinal cortex (ERC), subiculum (Sub), CA1, and CA3/Dentate Gyrus (CA3) in the resting states on the same person in 84 successive days. We used the anatomical connections

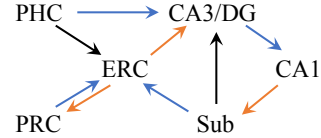

[BB08, ZHZ+17] as a reference. We applied both MC and IB on this dataset. We investigated all possible causal orders and found the one that minimizes the causal order indicator for MC and IB. After identifying the causal ordering, we estimated the causal coefficients $B$ on each domain separately with threshold $\alpha = 0.1$, and if an edge appears in more than 60% of all domains, we took this edge in the final graph. The recovered causal graph between the six regions is shown in the figure on the right. The black edges indicate edges, which are identified by both MC and IB methods. The blue edges are only identified by the MC method, and the orange edges are only identified by the IB method. The edges in the anatomical ground truth are as follows: PHC → ERC, PRC → ERC, ERC → CA3/DG, CA3/DG → CA1, CA1 → Sub, Sub → ERC, and ERC → CA1.

## 4 Conclusion

We studied causal structure learning from multi-domain observational data. We proposed methods based on the principle that in a causally sufficient system, the causal modules, as well as their included parameters, change independently across domains. The main idea in our approach does not require any type of invariance across the domains. We first introduced our methods on a linear system comprised of two variables, and then proposed efficient algorithms to generalize the idea to the case of multiple variables. We evaluated our method on both synthetic and real data. Our proposed methods are generally capable of identifying causal direction from fewer than ten domains, and when the invariance property holds, two domains are generally sufficient.

**Acknowledgments**

This work was supported in part by Army grant W911NF-15-1-0281 and NSF grant NSF CCF 1065022. This material is partially based upon work supported by United States Air Force under Contract No. FA8650-17-C-7715, by National Science Foundation under EAGER Grant No. IIS-1829681, and National Institutes of Health under Contract No. NIH-1R01EB022858-01, FAINR01EB022858, NIH-1R01LM012087, NIH-5U54HG008540-02, and FAIN-U54HG008540, and work funded and supported by the Department of Defense under Contract No. FA8702-15-D-0002 with Carnegie Mellon University for the operation of the Software Engineering Institute, a federally funded research and development center. Any opinions, findings, and conclusions or recommendations expressed in this material are those of the authors and do not necessarily reflect the views of the United States Air Force or the National Institutes of Health or the National Science Foundation. We thank Clark Glymour and Malcolm Forster for helpful discussions, and appreciate the comments from anonymous reviewers, which greatly helped to improve the paper.

## Footnotes

[2] We have provided an example in the Supplementary Materials to demonstrate why it is required to consider both orders $\pi_{X,-1}$ and $\pi_{X,-2}$.

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
