[Supplementary Material · MDCSLCR supplementary materials.pdf]

# Appendices

## A Proof of Theorem 1

For any parameter of interest, we denote its minimum value and its maximum value with subscripts $m$ and $M$, respectively, and its minimum value and its maximum value in the dataset with subscripts $\hat{m}$ and $\hat{M}$, respectively. Given dataset $\mathcal{D} = \{D^{(1)}, \cdots, D^{(d)}\}$, we first note that for $\gamma \in \{\sigma^2_{C|\emptyset}, \sigma^2_{E|\emptyset}\}$, as $d \to \infty$,

$$\gamma_{\hat{M}} \to \gamma_M, \qquad \gamma_{\hat{m}} \to \gamma_m. \tag{5}$$

We only prove the former, as the proof of the latter is similar.

For any $\epsilon > 0$, we define the event $F(\epsilon) = \{\gamma_M - \gamma_{\hat{M}} \geq \epsilon\}$. Also, we define

$$p^{(i)}(\epsilon) = P(\gamma_M - \epsilon \leq \gamma^{(i)} \leq \gamma_M).$$

We assume that the distribution of the parameters has a bounded support, and is positive on its support. Therefore, for all $i \in [d]$, $p^{(i)}(\epsilon) > 0$. Therefore, for all $\epsilon > 0$, we have

$$\lim_{d \to \infty} P(F(\epsilon)) = \lim_{d \to \infty} \prod_{i \in [d]} (1 - p^{(i)}(\epsilon)) = 0.$$

This implies that $\gamma_{\hat{M}} \to \gamma_M$ as $d \to \infty$.

Consider the logarithm of the regression parameters. For the causal direction we have

$$\log(Var(C)) = \log \sigma^2_{C|\emptyset} = \log \sigma^2_C, \qquad \log |\beta_{E|C}| = \log |a|, \qquad \log \sigma^2_{E|C} = \log \sigma^2_E.$$

For the reverse direction, we have

$$\log(Var(E)) = \log \sigma^2_{E|\emptyset} = \log(a^2 \sigma^2_C + \sigma^2_E),$$

$$\log |\beta_{C|E}| = \log(|a|\sigma^2_C) - \log \sigma^2_{E|\emptyset}, \qquad \log \sigma^2_{C|E} = \log(\sigma^2_C \sigma^2_E) - \log \sigma^2_{E|\emptyset}.$$

In the causal direction, due to independence of the cause mechanism from the effect mechanism, we have

$$V_1 := \mathbb{E}[\log |\beta_{E|C}| \mid_{\log \sigma^2_{C|\emptyset} = (\log \sigma^2_{C|\emptyset})_m}] = \mathbb{E}[\log |a| \mid_{\log \sigma^2_C = \log(\sigma^2_C)_m}] = \mathbb{E}[\log |a|],$$

$$V_2 := \mathbb{E}[\log |\beta_{E|C}| \mid_{\log \sigma^2_{C|\emptyset} = (\log \sigma^2_{C|\emptyset})_M}] = \mathbb{E}[\log |a| \mid_{\log \sigma^2_C = \log(\sigma^2_C)_M}] = \mathbb{E}[\log |a|],$$

$$V_3 := \mathbb{E}[\log \sigma^2_{E|C} \mid_{\log \sigma^2_{C|\emptyset} = (\log \sigma^2_{C|\emptyset})_m}] = \mathbb{E}[\log \sigma^2_E \mid_{\log \sigma^2_C = \log(\sigma^2_C)_m}] = \mathbb{E}[\log \sigma^2_E],$$

$$V_4 := \mathbb{E}[\log \sigma^2_{E|C} \mid_{\log \sigma^2_{C|\emptyset} = (\log \sigma^2_{C|\emptyset})_M}] = \mathbb{E}[\log \sigma^2_E \mid_{\log \sigma^2_C = \log(\sigma^2_C)_M}] = \mathbb{E}[\log \sigma^2_E],$$

and hence

$$V_5 := |V_2 - V_1| + |V_4 - V_3| = 0.$$

By (5), $\mathcal{T}^{IB}_{C \to E}(\mathcal{D}) \to V_5$. Therefore, $\mathcal{T}^{IB}_{C \to E}(\mathcal{D}) \to 0$ as $d \to \infty$.

Since $\log \sigma^2_{E|\emptyset}$ is a monotonically increasing function of $\sigma^2_C$, $|a|$, and $\sigma^2_E$, we have

$$\log \sigma^2_{E|\emptyset} = (\log \sigma^2_{E|\emptyset})_m \Leftrightarrow \sigma^2_C = (\sigma^2_C)_m, \ |a| = |a|_m, \ \sigma^2_E = (\sigma^2_E)_m,$$

and

$$\log \sigma^2_{E|\emptyset} = (\log \sigma^2_{E|\emptyset})_M \Leftrightarrow \sigma^2_C = (\sigma^2_C)_M, \ |a| = |a|_M, \ \sigma^2_E = (\sigma^2_E)_M.$$

Therefore, for the reverse direction we have

$$\tilde{V}_1 := \mathbb{E}[\log |\beta_{C|E}| \mid_{\log \sigma^2_{E|\emptyset} = (\log \sigma^2_{E|\emptyset})_m}] = \log(|a|_m (\sigma^2_C)_m) - \log(|a|^2_m (\sigma^2_C)_m + (\sigma^2_E)_m),$$

$$\tilde{V}_2 := \mathbb{E}[\log |\beta_{C|E}| \mid_{\log \sigma^2_{E|\emptyset} = (\log \sigma^2_{E|\emptyset})_M}] = \log(|a|_M (\sigma^2_C)_M) - \log(|a|^2_M (\sigma^2_C)_M + (\sigma^2_E)_M),$$

$$\tilde{V}_3 := \mathbb{E}[\log \sigma^2_{C|E} \mid_{\log \sigma^2_{E|\emptyset} = (\log \sigma^2_{E|\emptyset})_m}] = \log((\sigma^2_C)_m (\sigma^2_E)_m) - \log(|a|^2_m (\sigma^2_C)_m + (\sigma^2_E)_m),$$

$$\tilde{V}_4 := \mathbb{E}[\log \sigma^2_{C|E} \mid_{\log \sigma^2_{E|\emptyset} = (\log \sigma^2_{E|\emptyset})_M}] = \log((\sigma^2_C)_M (\sigma^2_E)_M) - \log(|a|^2_M (\sigma^2_C)_M + (\sigma^2_E)_M),$$

and hence

$$\tilde{V}_5 := |\tilde{V}_2 - \tilde{V}_1| + |\tilde{V}_4 - \tilde{V}_3| = |\log(\frac{|a|_M(\sigma_C^2)_M}{|a|_m(\sigma_C^2)_m}) + \log(\frac{|a|_m^2(\sigma_C^2)_m + (\sigma_E^2)_m}{|a|_M^2(\sigma_C^2)_M + (\sigma_E^2)_M})|$$
$$+ |\log(\frac{(\sigma_C^2)_M(\sigma_E^2)_M}{(\sigma_C^2)_m(\sigma_E^2)_m}) + \log(\frac{|a|_m^2(\sigma_C^2)_m + (\sigma_E^2)_m}{|a|_M^2(\sigma_C^2)_M + (\sigma_E^2)_M})|$$

If $\tilde{V}_5 = c$, Assumption 1 guarantees that $c \neq 0$. Also, by (5), $\mathcal{T}_{E \to C}^{IB}(\mathcal{D}) \to \tilde{V}_5$. Therefore, as $d \to \infty$, $\mathcal{T}_{E \to C}^{IB}(\mathcal{D}) \to c$ for some positive value $c$.

## B    Extension of the IB Method to Second-Order Test

For testing if $X$ is the cause of $Y$, let $\Gamma_{X \to Y} = \{|\beta_{Y|X}|, \sigma_{Y|X}^2\}$. For any parameter of interest, we denote its minimum and its maximum value in the dataset with subscripts $\hat{m}$ and $\hat{M}$, respectively. For inferring if $X$ is the cause of $Y$, let

$$\mathcal{I}^{IB2}(\gamma, \sigma_{X|\emptyset}^2) := \mathcal{I}^{IB}(\gamma, \sigma_{X|\emptyset}^2)$$
$$+ |Var(\log\gamma\,|_{\log(\sigma_{X|\emptyset}^2)=(\log(\sigma_{X|\emptyset}^2))_{\hat{M}}}) - Var(\log\gamma\,|_{\log(\sigma_{X|\emptyset}^2)=(\log(\sigma_{X|\emptyset}^2))_{\hat{m}}})|,$$

and according to (3), we define the causal direction indicator $\mathcal{T}_{X \to Y}^{IB2}(\mathcal{D}) := \sum_{\gamma \in \Gamma_{X \to Y}} \mathcal{I}^{IB2}(\gamma, \sigma_{X|\emptyset}^2)$. To see the causal relation between $X$ and $Y$, we calculate $\mathcal{T}_{X \to Y}^{IB2}(\mathcal{D})$ and $\mathcal{T}_{Y \to X}^{IB2}(\mathcal{D})$ and pick the direction which has the smaller value, i.e., $\arg\min_{\pi \in \{X \to Y, Y \to X\}} \mathcal{T}_\pi^{IB2}(\mathcal{D})$, as the true causal direction.

## C    Proof of Theorem 2

We first note that if for domains $D^{(i)}$ and $D^{(j)}$, $Q_{C \to E}^{(i,j)} \neq 0$, then at least one of the variables $a$, $\sigma_C^2$, or $\sigma_E^2$ has varied across the two domains and hence, by faithfulness assumption, $Q_{E \to C}^{(i,j)} = 3$. Noting that $0 \leq Q_{X \to Y}^{(i,j)} \leq 3$, this implies that

$$Q_{C \to E}^{(i,j)} \leq Q_{E \to C}^{(i,j)}, \qquad \forall i, j.$$

Summing up over $\{i, j\}$, it implies that $\mathcal{T}_{C \to E}^{MC}(\mathcal{D}) = \binom{d}{2}$, and hence $\mathcal{T}_{C \to E}^{MC}(\mathcal{D}) \geq \mathcal{T}_{E \to C}^{MC}(\mathcal{D})$.

If there exists a pair of domains $\{D^{(i)}, D^{(j)}\}$ for which $1 \leq Q_{C \to E}^{(i,j)} \leq 2$, then since $Q_{E \to C}^{(i,j)} = 3$, we have $Q_{C \to E}^{(i,j)} < Q_{E \to C}^{(i,j)}$. Therefore, $\mathcal{T}_{E \to C}^{MC}(\mathcal{D}) \leq \binom{d}{2} - 1$. Also, as mentioned earlier, $\mathcal{T}_{C \to E}^{MC}(\mathcal{D}) = \binom{d}{2}$. Therefore, in this case, we have $\mathcal{T}_{C \to E}^{MC}(\mathcal{D}) > \mathcal{T}_{E \to C}^{MC}(\mathcal{D})$.

## D    An Efficient Method for Estimating Coefficients and Noise Variances

From the SEM in equation (4), we have $X = (I - B)^{-\top} N$. Hence, the correlation matrix could be calculated as $\Sigma = \mathbb{E}[XX^\top] = (I - B)^{-\top}\Omega(I - B)^{-1}$, where $\Omega$ is a $p \times p$ diagonal matrix with values $\sigma_1^2, \cdots, \sigma_p^2$ on the diagonal. Therefore, the precision matrix, i.e., the inverse of the correlation matrix will be $\Theta = (I - B)\Omega^{-1}(I - B)^\top$. This expression could be used to obtain the elements of $\Theta$ as follows:

$$\begin{cases} \Theta_{i,i} = \sigma_i^{-2} + \sum_{j=1}^p B_{i,j}^2 \sigma_j^{-2} & \forall i \in [p], \\ \Theta_{i,j} = -B_{i,j}\sigma_j^{-2} - B_{j,i}\sigma_i^{-2} + \sum_{k=1}^p B_{i,k}B_{j,k}\sigma_k^{-2} & \forall i \neq j. \end{cases} \qquad (6)$$

Let $J(\cdot)$ be the operator that flips an input square matrix over its anti-diagonal. Define $G := J(\Theta)$. We calculate the LU-factorization of the matrix $G$, which is $G = LU$, where $L$ is a lower-triangular, and $U$ is an upper-triangular matrix, respectively. The procedure is shown in Algorithm 3. Due to the factorization step, the computational complexity of this algorithm is $\mathcal{O}(p^3)$. For input order $\pi$, the algorithm outputs $\hat{B}_\pi$ and $\hat{\Omega}_\pi$, which satisfy the following result.

---

**Algorithm 3** Parameter Estimator

**Input:** Order $\pi$.
Compute covariance matrix $\Sigma = \mathbb{E}[(\pi(1), \cdots, \pi(p))^\top (\pi(1), \cdots, \pi(p))]$, and $\Theta = \Sigma^{-1}$.
$G = J(\Theta)$, Perform LU-factorization on $G$: $G = LU$.
$\bar{\Omega} = $ Diagonal matrix with $\bar{\Omega}_{m,m} = 1/U_{m,m}$, for $1 \leq m \leq p$.
$\hat{B}_\pi = -J(\bar{\Omega}U) + I$, $\hat{\Omega}_\pi = J(\bar{\Omega})$.
**Output:** $\hat{\Omega}_\pi$ and $\hat{B}_\pi$.

---

**Theorem 7.** *For a given order $\pi$ on variables, for any $1 \leq m \leq p$, let $S_m := (\pi(1), \cdots, \pi(m-1))^\top$. In the population setup, for the matrices $\hat{B}_\pi$ and $\hat{\Omega}_\pi$ obtained from Algorithm 3, we have $(\hat{B}_\pi)_{(1:m-1),m} = \beta_{\pi(m)|S_m}$, and $(\hat{\Omega}_\pi)_{m,m} = \sigma^2_{\pi(m)|S_m}$, and zero elsewhere. Specifically, for a causal order, we have $\hat{B}_\pi = B$, and $\hat{\Omega}_\pi = \Omega$ up to permutation.*

Theorem 7 shows the connection between the covariance matrix and the adjacency matrix of the causal network.

*Proof.* Consider the last variable in the given order, i.e., $\pi(p)$. According to Algorithm 3, for any $1 \leq m \leq p - 1$, we have

$$\hat{B}_{m,p} = -\frac{U_{1,m}}{U_{1,1}} = -\frac{L_{1,\cdot}^{-1}G_{\cdot,m}}{L_{1,\cdot}^{-1}G_{\cdot,1}} = -\frac{G_{1,m}}{G_{1,1}} = -\frac{\Theta_{m,p}}{\Theta_{p,p}}.$$

As observed in [Pou11], this value is equal to the element of $\beta_{\pi(p)|S_p}$ corresponding to $\pi(m)$. Therefore,

$$\hat{B}_{(1:p-1),p} = \beta_{\pi(p)|S_p}.$$

Similarly, for any $1 \leq m \leq p - 1$, we have

$$\hat{\Omega}_{p,p} = \bar{\Omega}_{1,1} = \frac{1}{U_{1,1}} = \frac{1}{L_{1,\cdot}^{-1}G_{\cdot,1}} = \frac{1}{G_{1,1}} = \frac{1}{\Theta_{p,p}},$$

which is equal to $\sigma^2_{\pi(p)|S_p}$.

Next, we show that in the elimination step in Algorithm 3, we remove the linear effect of $\pi(p)$ from all preceding variables in the given order, and an inductive argument over the variables eliminated in each next round gives the desired result: In the elimination, we add the vector $-\frac{G_{m,1}}{G_{1,1}}G_{1,\cdot}$ to the $m$-th row of $G$, that is for all $k$, $G_{m,k}$ will be updated as follows:

$$(G_{m,k})_{new} = (G_{m,k})_{old} - \frac{G_{m,1}}{G_{1,1}}G_{k,1}$$

$$(G_{m,k})_{new} = (G_{m,k})_{old} - \frac{G_{m,1}}{G_{1,1}}\frac{G_{k,1}}{G_{1,1}}G_{1,1}$$

$$(G_{m,k})_{new} = (G_{m,k})_{old} - [\beta_{\pi(p)|S_p}]_m [\beta_{\pi(p)|S_p}]_k \sigma^{-2}_{\pi(p)|S_p}.$$

This removes the linear effect of $\pi(p)$ from $\pi(m)$. Specifically, if the order is consistent, we are eliminating $B^2_{m,p}\sigma^{-2}_p$ from $\Theta_{m,m}$, and $B_{m,p}B_{k,p}\sigma^{-2}_p$ from $\Theta_{m,k}$, which are terms related to $\pi(p)$ as seen in equation (6). $\qquad\square$

---

**Remark 2.** *We only need to calculate $\Theta$ in one domain and for one order: There exist methods in the literature, e.g., [XG16], for calculating the difference between two precision matrices, i.e., $\Delta^{(i,j)} = \Theta^{(j)} - \Theta^{(i)}$ directly. Therefore, we can calculate the inverse only in one domain, say, domain $D^{(i)}$ (preferably the one in which we have the largest set of samples), and then for any other domain $j$, obtain $\Theta^{(j)} = \Theta^{(i)} + \Delta^{(i,j)}$ Also, as seen in equation (6) the elements of $\Theta$ permute according to an arbitrary ordering and recalculation is not required.*

# E  Proof of Theorem 3

The proof has the same rationale as Theorem 1 and we avoid repeating the details. Due to the independence among the parameters of the system, an argument exactly the same as the one in Theorem 1 shows that $\mathcal{T}_{\pi^*}^{eIB}(\mathcal{D}) \to 0$.

In order $\pi'$, we find the last variable which is positioned after at least one if its children, Let this variable be $X_u$ and a child that comes before it in the order be $X_v$. Let $\Omega_u$ and $\Omega_v$ be the elements of $\hat{\Omega}_{\pi'}$ corresponding to $X_u$ and $X_v$, respectively. Since $X_u$ is missing when regressing $X_v$, $\Omega_v$ will be an increasing function of $B_{u,v}$ and $\sigma_u^2$. Therefore, for the maximum value of $\Omega_v$, $B_{u,v}$ and $\sigma_u^2$ are fixed to $(B_{u,v})_M$ and $(\sigma_u^2)_M$, respectively, and other parameters of the system involved in $\Omega_v$ are also fixed. Similarly, for the minimum value of $\Omega_v$, $B_{u,v}$ and $\sigma_u^2$ are fixed to $(B_{u,v})_m$ and $(\sigma_u^2)_m$, respectively, and other parameters of the system involved in $\Omega_v$ are also fixed. By [Pou11], $\Omega_u$ is a rational function of parameters of the system, including $B_{u,v}$ and $\sigma_u^2$. When taking the expected value of $\Omega_u$ conditioned on maximum and minimum values of $\Omega_v$, the parameters involved in $\Omega_v$ are all fixed, and the parameters not involved are independent of $\Omega_v$. Therefore, since $B_{u,v}$ and $\sigma_u^2$ are taking two distinct values in the maximum and minimum, by the faithfulness assumption, the expected value conditioned on minimum and maximum of $\Omega_v$ will be different. Therefore, since $\mathcal{T}_{\pi'}^{eIB}(\mathcal{D})$ is the sum of absolute values, including the absolute value of the difference of the aforementioned expected values, it will be bounded away from zero. The argument for extending this result to the case with samples and letting $d \to \infty$ is the same as Theorem 1.

# F  Proof of Theorem 4

Similar to the proof of Theorems 1 and 3 and , the claim $\mathcal{T}_{X_s,-1}^{IB}(\mathcal{D}) \to 0$ is due to the independence among the parameters of the system. Also, the claim $\mathcal{T}_{X_v,-1}^{IB}(\mathcal{D}) \to c$ could be concluded from the proof of Theorem 3, as the last variable which is positioned after at least one if its children would be $X_v$ itself.

# G  Proof of Theorem 5

We relabel variables according to $\pi^*$ to have $\pi^*(i) = X_i$, that is, in the causal order, any variable with smaller label proceeds variables with larger labels. Since $\pi^*$ is causal, $\hat{B}_{\pi^*} = B$, and $\hat{\Omega}_{\pi^*} = \Omega$. Therefore, $\Gamma'_{\pi^*}$ is exactly the set of parameters of the system. Therefore, for a pair of domains $\{D^{(i)}, D^{(j)}\}$, $Q_{\pi^*}^{(i,j)}$ denotes exactly how many of the parameters of the system have changed across domains $D^{(i)}$ and $D^{(j)}$.

On the other hand, since $\pi'$ is not causal, there exists parent variables who are regressed on their children, and hence, the corresponding elements of $\hat{B}_{\pi'}$ and $\hat{\Omega}_{\pi'}$ will be functions of more that one parameter of the system. Therefore, by faithfulness assumption, they will vary by a change in any of the involved parameters across any two domains $D^{(i)}$ and $D^{(j)}$. Therefore, an argument similar to the one in the proof of Theorem 2 implies that

$$\mathcal{T}_{\pi^*}(\mathcal{D}) \geq \mathcal{T}_{\pi'}(\mathcal{D}).$$

Also, since $\pi'$ is not causal, there exists indices $i$ and $j$, such that $(X_i, X_j) \in A$ (recall that $A$ is the set of directed edges), but $(\pi')^{-1}(X_i) > (\pi')^{-1}(X_j)$. Having $\pi'$ as the order, we regress $X_i$ on a set $S$ including $X_j$. We denote the coefficient corresponding to $X_j$ by $\beta$, and the variance of the residual of the regression by $\sigma^2$.

First, we note that $\beta$ will be non-zero, as $X_j$ is in the Markov blanket of $X_i$. Applying the result of [Pou11], $\beta$ and $\sigma^2$ can be written as follows

$$\beta = \frac{\tilde{B}_{i,j}\tilde{\sigma}_j^{-2} - \sum_{k:X_k \in S} \tilde{B}_{i,k}\tilde{B}_{j,k}\tilde{\sigma}_k^{-2}}{\tilde{\sigma}_i^{-2} + \sum_{k:X_k \in S} \tilde{B}_{i,k}^2 \tilde{\sigma}_k^{-2}},$$

and

$$\sigma^2 = (\tilde{\sigma}_i^{-2} + \sum_{k:X_k \in S} \tilde{B}_{i,k}^2 \tilde{\sigma}_k^{-2})^{-1},$$

where $\tilde{\sigma}_i^2$ and $\tilde{B}_{i,j}$ are the variance of the residual and the coefficient in the subgraph induced on $\{X_i\} \cup S$. Due to the faithfulness assumption, the correlations will not be cancelled out, and hence, $\beta$ and $\sigma^2$ depend on $\tilde{\sigma}_i^2$ and $\tilde{B}_{i,j}$, which in turn depend on $\sigma_i^2$ and $B_{i,j}$. Therefore, if , say, $B_{i,j}$ remains fixed and $\sigma_i^2$ varies across two domains $D^{(i)}$ and $D^{(j)}$, then similar to the proof of Theorem 2, we will have

$$\mathcal{T}_{\pi^*}(\mathcal{D}) > \mathcal{T}_{\pi'}(\mathcal{D}).$$

## H  An Example For Requirement of considering both orders $\pi_{X,-1}$ and $\pi_{X,-2}$ in Algorithm 2

Suppose the ground truth structure is $X_1 \rightarrow X_2 \rightarrow X_3 \rightarrow X_4 \rightarrow X_5$, and suppose we start with initial ordering $\pi_t = \{1, 5, 4, 2, 3\}$. If Algorithm 2 does not consider $\pi_{X,-2}$, the following can happen:

**Round 1:** Algorithm 2 forms $\Pi_{X_1} = \{\pi_t, \pi_{X_1,-1}\}$. We have $\pi_t = \arg\max_{\pi \in \Pi_{X_1}} \mathcal{T}_\pi^{MC}(\mathcal{D})$. Therefore, the ordering will not change.

**Round 2:** Algorithm 2 forms $\Pi_{X_5} = \{\pi_t, \pi_{X_5,-1}\}$. We have $\pi_{X_5,-1} = \arg\max_{\pi \in \Pi_{X_5}} \mathcal{T}_\pi^{MC}(\mathcal{D})$. Therefore, the ordering will change to $\pi_t = \{1, 4, 2, 3, 5\}$.

**Round 3:** Algorithm 2 forms $\Pi_{X_4} = \{\pi_t, \pi_{X_4,-1}\}$. We have $\pi_{X_4,-1} = \arg\max_{\pi \in \Pi_{X_4}} \mathcal{T}_\pi^{MC}(\mathcal{D})$. Therefore, the ordering will change to $\pi_t = \{1, 2, 3, 5, 4\}$.

**Round 4:** Algorithm 2 forms $\Pi_{X_2} = \{\pi_t, \pi_{X_2,-1}\}$. We have $\pi_t = \arg\max_{\pi \in \Pi_{X_2}} \mathcal{T}_\pi^{MC}(\mathcal{D})$. Therefore, the ordering will not change.

**Round 5:** Algorithm 2 forms $\Pi_{X_3} = \{\pi_t, \pi_{X_3,-1}\}$. We have $\pi_t = \arg\max_{\pi \in \Pi_{X_3}} \mathcal{T}_\pi^{MC}(\mathcal{D})$. Therefore, the ordering will not change.

Therefore Algorithm 2 outputs $\pi_t(-1) = 4$ as a sink variable while it is not a sink.

## I  Proof of Theorem 6

Since $X_s$ is a sink variable, by moving it to the last position in the order, none of its ancestors will be regressed on it, and hence, this move minimizes the dependencies among estimated regression parameters, which in turn minimizes the number of varying parameters. Therefore, for all $\pi \in \Pi_{X_s}$, $\mathcal{T}_{\pi_{X_s,-1}}^{MC}(\mathcal{D}) \geq \mathcal{T}_\pi^{MC}(\mathcal{D})$.

Suppose in the initial order $\pi_t$, there is a sink variable $X_s$ as $\pi_t(-1)$. Then for any other variable $X_v$, moving it to $\pi_t(-1)$ either increases the dependencies or if , say, $X_v$ is also a sink variable, will not change it. Therefore, based on our prioritization, $X_s$ will remain in position $\pi_t(-1)$ until the end of the round. If in the initial order there is a non-sink variable as $\pi_t(-1)$, when we are checking its source ancestor $X_s$, since there exists a pair of domains across which at least 1 and at most 2 of variables $Var(X_v)$, $B_{v,s}$, $\sigma_s^2$ varies, moving $X_s$ below $X_v$ will increase the value of the causal order indicator; that is, for all $\pi \in \Pi_{X_s}$, $\mathcal{T}_{\pi_{X_s,-1}}^{MC}(\mathcal{D}) > \mathcal{T}_\pi^{MC}(\mathcal{D})$. Therefore, $X_s$ will move to the bottom of the order, and similar to the previous case, it will remain at that position until the end of that round. Therefore, in either case, at the end of round, $\pi_t(-1)$ will be a sink variable.