[Reviews · NeurIPS 2018]

Reviewer 1



The authors leverage on data recorded in multiple domains with changing parameters of a linear Gaussian models to learn causal direction and relations beyond Markov equivalence class. The paper is clearly written and includes good synthetic data simulations. The result is theoretically interesting. The method seems to need a lot of samples and domains that may not be available in real cases. The authors present different options to improve the methods in this respect. In the abstract the authors claim that they “do not assume any invariance of the causal process across the domains”: But firstly, it seems the causal structure stays the same. What happens with domains with intervention experiments that change the causal structure? Also, without any invariance assumption the causal direction may change from domain to domain. Secondly, the authors assume that the means (of the noises) stays at zero in all domains. This seems odd here, although the authors claim that they make this “without loss of generality” (which is of course true for a single domain system). I think the paper should absolutely also deal with the change of the mean in the different domains, since this would be very much expected in any real data. Changes in the mean should be noted in eqs 1 & 2. (Note that intervention may also change the mean, so I guess the current model does not allow for interventions.) The simulations are in need of a baseline: since the parameters of the Gaussian model change, the pooled data is possibly linear non-Gaussian, or non-linear non-Gaussian ANM. So it would make a lot of sense to plot LiNGAM and ANM for the pooled data as a baseline. The paper could cite the work on transportability by Bareinboim & Pearl where similarly parameters change across domain, although the focus there is inference not discovery. Also the work of Kummerfield and Danks (NIPS) on online varying causal structure seems relevant here. AFTER AUTHOR REBUTTAL: -Still: mean is a parameter similarly as the variance and its changes could be used. If not, longer discussion is needed on this and why mean is not analyzed similarly as variance. -"LiNGAM cannot be used", you can use it, just input the data and see the result. The result may be poor but this is then good to your algorithm. Some baseline would be good.

Reviewer 2



This paper proposes an approach to learn causal directionality -- first in the two-variable case, then generalized to DAGs, in the settings where multiple datasets of the same system are recorded across different "domains" (could be different time points, or different measurement instruments for example). The approach is based on an assumption called the "independent change condition" (ICC): for cause C and effect E, P(C|E) and P(E) should change independently across different domains. Further, linearity is assumed. Under these assumptions, the authors develop interesting algorithms that first identify causal directions in the two-variable case, which is then generalized to structure learning. Importantly these algorithms could be used with a partial result from another structure learning method, such as PC, as input to identify further possible orientations. Strong aspects of the paper are: - The idea is innovative (the ICC concept itself is not new, but the algorithmic ideas appear to be) and leads to an interesting class of methods, which might be extensible further. - The paper is well-written and clear (up to a few issues noted below). - Due attention is given to the efficiency of the statistical testing procedures that underpin the algorithm. Two variants are presented in Section 3 that substantially increase power and reduce computation efforts compared to "naive" approaches. I emphasize this because structure learning papers sometimes tend to neglect the issues surrounding statistical testing. Points of criticism are: - Significance: I frankly speaking do not understand why we the ICC would be a good assumption. In the article, only two literature references are given (which by the way are hard to locate due to errors in the bibliography: DJM+10 appears to have a wrong author list and ZHZ+17 has a wrong title). I would have liked more motivation or perhaps a real-world example explaining why this would be a good assumption. If we allow all parameters to change across domains, why would specific parameters change only independently? - Clarity: I had trouble with Definition 3, which is central for understanding Theorem 4. What does "ordering" mean here precisely? It cannot be just a permutation of the variables, as one might suspect when looking at Theorem 4, because a permutation has no edges. From the examples given it seems that "ordering" means instead "DAG" but that cannot be true either since otherwise all DAGs with no edges would be consistent with all other DAGs according to the definition. My working assumption was that order here means in fact a "connected partial order", in which case Theorem 4 appears to hold, but I might be misunderstanding this. It would be very useful if the authors could clarify this. - Quality: The paper would benefit from a real-world example to demonstrate the applicability of the proposed techniques. Only simulated data is being used at the moment. Minor points: - The paper should have a conclusion, even if it's only brief. - Assumption 1: I was wondering here why this is called "Faithfulness". You do explain this later after Assumption 2 (for the network case) and I'd suggest moving up this explanation. - Algorithm 1: I was wondering if it is really so important in which way you compute these regression coefficients and residual variances . Wouldn't this provide the same results as, for instance, simply running the regression of each variable on each other variable with higher order? I understand that you need to explain this somewhere to get to precise algorithm runtimes but lines 215 to 224 seemed to interrupt the flow of exposition a little and seemed to get in the way of your explanation of the actual structure learning approach. Perhaps it could just be mentioned that there's a way to compute all required parameters in time O(p^3) and put the details into supplement. = After author feedback = Thank you for your responses. A clarification of the "ordering" concept in the final version would indeed be greatly appreciated.

Reviewer 3



The paper studies causal discovery in linear systems given a set of purely observational data D^{(1)}, D^{(2)}, … in multiple domains such that all of the datasets D^{(i)} are generated from the same causal structure but the causal parameters may vary. The proposed approach is based on the principle that “P(cause) and P(effect |cause), as well as their included parameters, change independently across domains”. Therefore in a linear causal model C->E given by C= N_C, E=aC+N_E, variance \sigma^2_C changes independently from (a, \sigma^2_E). If we assume these causal parameters are random variables and the causal parameter values across domains are i.i.d. samples, then we could identify the causal direction by testing the independence between \sigma^2_C and (a, \sigma^2_E) if enough number of domains are available. The paper first presents two test methods for determining the causal direction between two (unconfounded) variables and then extends the methods to causal network structure learning algorithms. The proposed causal discovery method is novel as far as I know. The paper contains valuable contributions to the field that merit publication. The main weakness of the work is the strong perhaps not practical assumptions that data from many domains are available and that the causal parameters across domains are random samples. The experimental evaluations are done by simulating data under these assumptions. It is not clear how the proposed methods would perform under real world situations. It would be desirable the paper could motivate the work with possible applications and evaluate the methods in more realistic settings. The paper is mostly clearly written. One key idea I couldn’t understand is the test statistic described in line 162-167. What is the meaning of this test statistic and how exactly is the expression computed? The paper mentioned the motivation for the proposed IB and MC methods is that “a general non-parametric independence test may not be efficient.” Should you actually try a general independence test and compare with the proposed IB and MC methods as a baseline? -Line 95, “form” should be “from”. After author feedback: - I don’t think the assumption that the causal parameters across domains are random samples is a consequence of not having latent confounders in the system. -It looks to me the performance of the proposed method and the relative performances of MC, IB, and HSIC test would strongly depend on how the parameters are actually changing (or not changing) across domains. In reality, if we don't know how parameters are changing, how much can we trust the output or the output by which test methods should we trust more?